# Using DHIS2 routine data for health system preparedness in resource-limited settings: A Bayesian predictive approach in Bangladesh

Toufiq Hassan Shawon[1◉], Ema Akter[2◉], Bibek Ahamed[2], Tazreen Ahmed[1], Romana Afroz Lubna[1], Animesh Biswas[1], Tasnu Ara[2], Ridwana Maher Manna[2], Pradip Chandra[2], Nasimul Gani Usmani[2], Md. Alamgir Hossain[2], Md. Shahidul Islam[2], S. M. Hasibul Islam[2], Sultan Mahmud[2], Abu Bakkar Siddique[2], Shafiqul Ameen[2], Shabnam Mostari[3], Mohammad Sohel Shomik[4], Emily Wilson[5], Nadia Akseer[5], Ahmed Ehsanur Rahman[2], Shams El Arifeen[2‡], Agbessi Amouzou[5‡], Aniqa Tasnim Hossain[2‡]*

1 Management Information System, Directorate General of Health Services, Dhaka, Bangladesh, 2 Maternal and Child Health Division, International Centre for Diarrhoeal Disease Research, Dhaka, Bangladesh, 3 Aspire to Innovate (a2i) Programme, Dhaka, Bangladesh, 4 Nutrition Research Division, International Centre for Diarrhoeal Disease Research, Dhaka, Bangladesh, 5 Global Disease Epidemiology and Control, Johns Hopkins Bloomberg School of Public Health, Baltimore, Maryland, United States of America

◉ These authors are joint first authors.
‡ SEA, AA and ATH are joint senior authors
* aniqa.hossain@icddrb.org

## Abstract

Health systems in low- and middle-income countries (LMICs) like Bangladesh face persistent challenges in delivering timely and equitable care, often exacerbated by poor planning and inefficient resource allocation. Forecasting service utilization using routine health data can support more responsive and data-driven health system planning, yet such approaches remain under utilized in Bangladesh. By analyzing service utilization trends and projecting future service volume at national and regional levels, we aim to improve region-specific health planning. This can promote more efficient and equitable service provision. We analyzed monthly routine health service data reported into the District Health Information Software 2 (DHIS2) platform between January 2021 and March 2025 in Bangladesh. We examined key indicators across maternal, newborn, child and hospital-based services. Bayesian log-linear Poisson regression models, adjusted for seasonality and autocorrelation, were applied to forecast service utilization for the final nine months of 2025 and all of 2026. Relative changes in 2025 and 2026 were calculated using 2024 as the reference year. The analysis revealed rising trends across most service areas relative to 2024 levels. Kangaroo Mother Care (KMC) has the highest projected expansion, with coverage forecast to rise by over 75% by 2026. Over the same time period, outpatient visits and pneumonia treatment are also expected to increase by about 30%. More moderate increases are seen in low birth weight (LBW) deliveries, cesarean sections, and

**Data availability statement:** The data underlying this study are from the District Health Information System 2 (DHIS2) in Bangladesh. These data are not publicly available but can be accessed upon reasonable request and with permission from the Directorate General of Health Services (DGHS), Ministry of Health and Family Welfare. Access to this data will be granted to qualified researchers upon reasonable request in DGHS (contact: dghs.portal.gov.bd). We are also able to share the anonymised data which was used in this analysis.

**Funding:** This work was guaranteed by the Gates Foundation through the United States Fund for UNICEF under Phase III of the Countdown to 2030 initiative (grant number INV-042414 to ATH). The funder had no role in study design, data collection, and analysis. The Gates Foundation had no role in publication.

**Competing interests:** The authors have declared that no competing interests exist.

normal deliveries. Notable regional disparities persist, with Dhaka and Chittagong showing the highest service utilization, while Barishal and Sylhet consistently report the lowest levels. Bangladesh's health system must prepare for increasing service utilization across all service categories. Forecasting using DHIS2 data supports for proactive planning and equitable resource allocation. Strategic investments in infrastructure, workforce, and data-driven planning are essential for building a resilient health system.

## Introduction

Health systems in low- and middle-income countries (LMICs) frequently face significant challenges in delivering timely, equitable and high-quality healthcare services [1]. These challenges are often exacerbated by constrained resources, weak infrastructure, workforce shortages and inefficient planning. The timely delivery of treatment is directly affected by the inefficient use of medical resources, which is an increasing problem for healthcare services [2]. According to research, a lack of access to high-quality care results in the deaths of around 8 million individuals annually [3]. In addition, millions of pregnant and postpartum women in low-income countries experience life-threatening complications and severe health conditions [4]. Each year, around 300,000 women die due to pregnancy or childbirth, with 94% of these deaths occurring in LMICs, especially in South Asia [5]. Most of these deaths are preventable with timely and appropriate care [5].

In Bangladesh, limited access to health services and under utilization of available care worsen the situation [6]. Bangladesh's health system faces multiple challenges, including poor intersectoral coordination, a shortage of qualified health professionals, limited budgetary resources, high out-of-pocket healthcare costs and significant inequities in service access [7]. The country has only 0.83 physicians per 1,000 population—far below the WHO-recommended minimum of 1.5 per 1,000 required to meet the Sustainable Development Goals (SDGs) [8]. Moreover, about 50% of medical equipment in public health facilities remains unused, with only 17% in working condition and the rest either non-functional or never installed [9]. Due to the scarcity of physicians, health workers and services, Bangladesh struggles to ensure even basic healthcare for all citizens [10].

Resource allocation decisions often lack data-driven insights, causing inefficiencies and service gaps. For managers and policymakers, the task of allocating scarce healthcare resources has been more challenging [11,12]. The rate of population growth significantly effects to planning for healthcare facilities and community health services [13,14]. According to one report, Bangladesh's healthcare system faces significant challenges due to the country's rapid population increase [15]. These inefficiencies are rooted not only in resource scarcity but also in insufficient planning and a lack of reliable predictive tools. To the best of our knowledge, no study to date has attempted to forecast health service utilization in Bangladesh. Accurate forecasts of patient visits and service use are essential for rational planning and resource

allocation. Reliable forecasting systems can issue early warnings of patient overflows, help optimize staffing and supplies and support the equitable and cost-effective distribution of medical resources.

Routine health information systems, such as the District Health Information Software 2 (DHIS2), offer a promising opportunity to strengthen health system preparedness and planning [16]. Its routinely collected service data, available at the facility level, can provide near real-time insights into service utilization patterns, seasonal fluctuations and emerging demands [16]. Timeliness of data input varied from 34% to 95%, while completeness varied from 50% to 98% across numerous DHIS2 datasets. The information is typically regarded as trustworthy and less susceptible to systematic bias because these records are kept by qualified medical experts [17]. Despite its potential, this rich data source remains underutilized for proactive forecasting and strategic planning.

In a country like Bangladesh, characterized by diverse geographic, economic and healthcare infrastructure across its eight administrative divisions—understanding regional variation in service utilization is critical. These regional differences may lead to distinct patterns in service utilization, posing unique challenges for health service delivery. Examining these differences is essential for understanding health system priorities and tailoring preparedness strategies to local contexts.

This study applies predictive modeling techniques to routine health data from DHIS2 to estimate service demand at the divisional level and explore variations in health service utilization and need across Bangladesh's eight divisions. Our goal is to generate actionable evidence to support better health system planning and preparedness. By identifying trends and forecasting demand, we aim to inform region-specific policy decisions and contribute to more efficient and equitable health service delivery.

## Methods

### Study settings

Situated in South Asia, Bangladesh is characterized by a large and diverse population. As reported in the 2022 Population and Housing Census, the country's population is estimated at around 165 million and the population growth rate is 1.22%, placing it among the world's most densely populated nations, with approximately 1,119 individuals living per square kilometer [18]. The country is organized into eight major administrative divisions: Barishal, Chittagong, Dhaka, Khulna, Rajshahi, Rangpur, Mymensingh and Sylhet.

### Data source

This study uses healthcare service use data collected from medical facilities across Bangladesh and reported in the DHIS-2. The DHIS2, a platform recommended by the World Health Organization (WHO) for managing health information in low resources countries. Since 2009, DHIS2 has been introduced into Bangladesh's national Health Management Information System (HMIS) to facilitate routine evaluation and monitoring of the provision of healthcare services. During regular service delivery, healthcare practitioners document health services in real time, which are then uploaded daily and in some cases monthly to the DHIS2 platform and recorded into facility registers [19]. This system enables evidence-based planning and decision-making and makes regular health service data easily accessible in a timely manner [17]. We extracted monthly service utilization data from January 2021 to March 2025 at division level for this analysis.

### Indicator

We examined several key service domains such as maternal health (number of normal deliveries, number of cesarean section), newborn health (total no. of low birth weight babies (less than 2500 grams), babies born in facility receiving kangaroo mother care (KMC), child health (number of children with pneumonia, diarrhoea with severe dehydration, measles, third dose of pentavalent vaccine) and hospital-based services (total outdoor patients, total admission patients, under-5 outdoor patients and under-5 admission patients).

## Analysis approach

Following prior research, we used a Bayesian log-linear Poisson model to forecast future health service use [20]. Historical monthly data used covered from January 2021 to March 2025 and projections were made for April-December 2025 and the year 2026. Before performing analysis, the dataset was examined for irregularities, missing values and outliers. Each study variable underwent completeness tests and no missing values were found. S5 File now includes the variable description table. In our Bayesian framework, we used Gaussian prior distributions for the time covariate and the intercept. Initially, we ran a linear regression model using time as the explanatory variable and monthly counts as the outcome. The previous distributions were informed by the resulting coefficient estimates and their accompanying standard deviations, where the standard deviations represented prior uncertainty and the estimates defined the prior means. We used the autocorrelation function (ACF) to assess the autocorrelation as the data was time-series, an autoregressive model of order one (AR(1)) was incorporated. We included autoregressive terms to account for temporal dependencies where autocorrelation was found. In order to assure correct modeling of periodic variations, we additionally adjusted for recurrent seasonal patterns to address seasonal variance in the monthly data. The INLA "seasonal" model used a cyclic seasonal component with a 12-month timeframe to model seasonality for monthly data.

The autocorrelation parameter ($\rho$) and precision ($\tau$), for which penalized complexity (PC) priors were provided, were the hyperparameters defining the temporal effects in the INLA framework. The model was predicated on cyclic seasonality, stationarity of the AR(1) process, and Gaussian observation errors. We applied posterior predictive checks, an ordinary and reliable diagnostic in Bayesian analysis, to evaluate the model's suitability (S4 File). The projections were conservative and assumed that government and partners' service demand generation and access activities will continue as is in the future.

During the last nine months of 2025 and each month of 2026, we recorded the mean and 95% credible interval (CrI) for the expected number of individuals accessing health services. In addition, we calculated the monthly averages for the period from 2024 to 2026 and assessed the relative change in the predicted values for the final nine months of 2025 and all of 2026, using the monthly averages of observed data from 2024 as a reference. Using the Integrated Nested Laplace Approximation (INLA) method, all models were implemented using R's "INLA" package (version 4.3.3; R Core Team, Vienna, Austria).

## Result

In Bangladesh, both normal and C-section deliveries have been gradually increasing. A clear seasonal pattern is observed, with both types of births peaking between September and December (S1 File). Projections indicate that the number of normal deliveries will rise by approximately 4% in 2025 and 12% in 2026 compared to 2024 (Fig 1). Normal deliveries are on the rise in Dhaka, Chittagong, Mymensingh and Sylhet, whereas declining trends are observed in Barishal, Khulna and Rangpur.

C-section deliveries are expected to increase by 6% in 2025 and 10% in 2026 (Fig 1), with divisional variations: upward trends are noted in Dhaka, Chittagong, Mymensingh and Rajshahi, while Barishal, Khulna, Sylhet and Rangpur are experiencing declines (S1 File).

Facility-based delivery of babies receiving Kangaroo Mother Care (KMC) is also growing. The number of newborns receiving KMC is projected to rise significantly, by around 20% in 2025 and 74% in 2026 compared to 2024 (Fig 1). Except for Khulna, all divisions showed an upward trend in KMC coverage.

Conversely, the number of low birth weight (LBW) newborns showed a modest increase, with a distinct seasonal peak from July to December (S1 File). Additionally, LBW births are expected to increase by 10% in 2025 and 25% in 2026, with most divisions showing a rising trend, except for Barishal, which is the only division with a projected decrease (Fig 1).

Our study observed a slight upward trend in diarrhea cases over time and a more notable increase in the number of under-five children receiving treatment for pneumonia. Diarrhea cases are projected to increase by 1% in 2025 and 5%

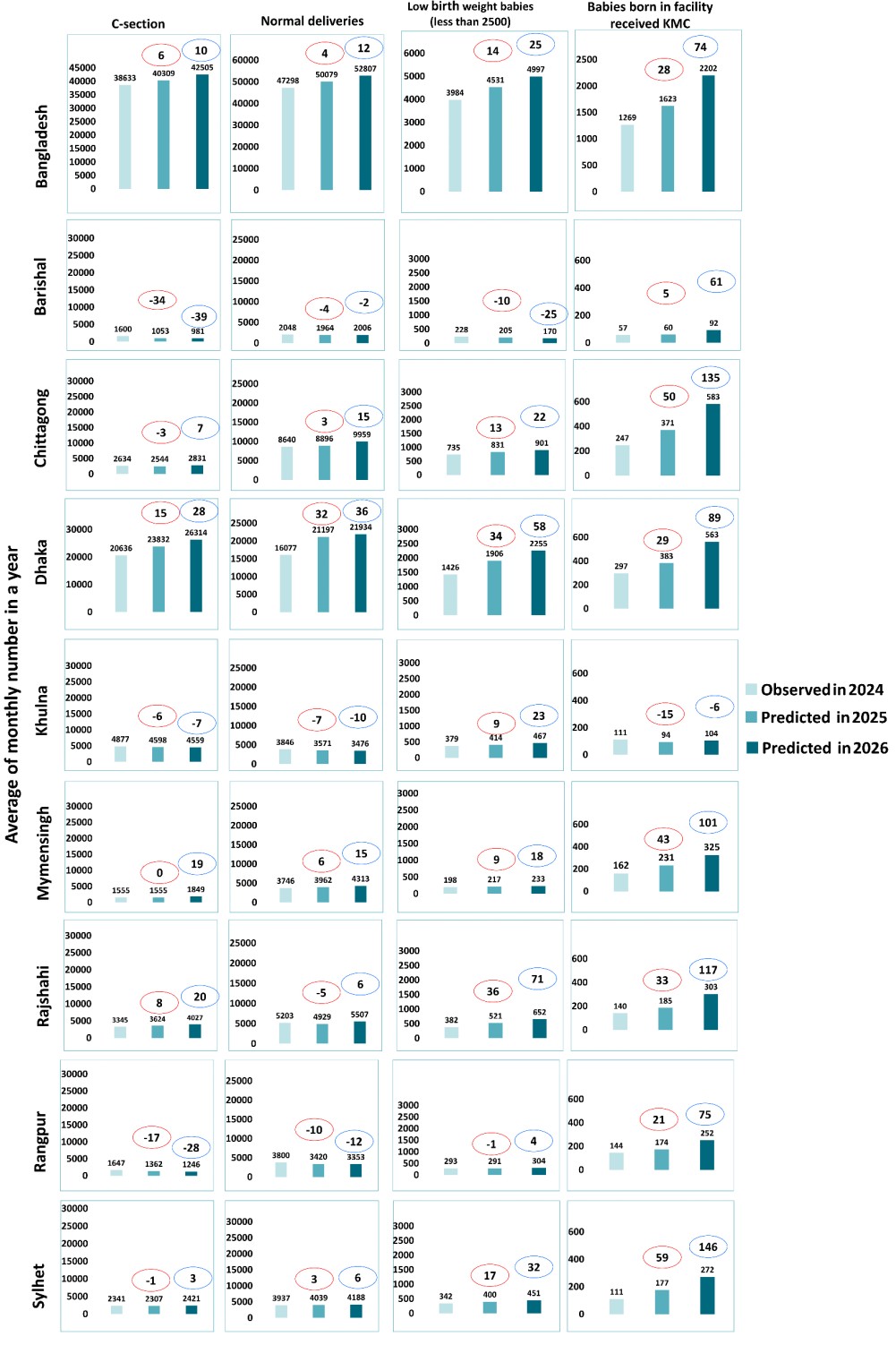

Red circles indicate the projected relative percentage change from 2024 to 2025.
Blue circles indicate the projected relative percentage change from 2024 to 2026.

**Fig 1. Relative change in maternal and newborn health service utilization during 2025 and 2026 compared to 2024.** Bars represent number of maternal and newborn service utilization per year.

in 2026 compared to 2024 (Fig 2). While Barishal, Chittagong and Sylhet are expected to see declines in diarrheal cases, Dhaka is projected to report the highest number and Sylhet the lowest (S2 File).

In contrast, pneumonia cases are expected to rise substantially, with projected increases of 9% in 2025 and 31% in 2026. Rangpur has the fewest children receiving treatment for pneumonia, while Dhaka reports the highest numbers. All divisions show an upward trend, but Dhaka is anticipated to experience the sharpest increase, over 50% more cases by 2026 compared to 2024 (Fig 2). Distinct seasonal patterns emerged: pneumonia treatments typically rose between October and December, while diarrhea cases peaked from October to November (S2 File).

Regarding immunization, Penta-3 vaccine coverage is expected to increase by 3% in 2025 and 1% in 2026 compared to 2024. Similarly, measles vaccination coverage is projected to rise by 2% in 2025 and 4% in 2026. Across divisions, vaccination rates exhibit minor fluctuations, with no significant changes observed (Fig 2).

There is a clear upward trend in hospital admissions, under-five admissions, outpatient visits and under-five outpatient visits, with the most pronounced seasonal peaks occurring between July and December. Based on data up to 2024, projections indicate that the utilization of health services will continue to increase in the coming years (S3 File).

Specifically, total hospital admissions are expected to rise by 8% in 2025 and 27% in 2026 compared to 2024. Admissions for children under five are projected to increase by 7% in 2025 and 28% in 2026. Among the divisions, Sylhet and Barishal have the lowest number of admissions, while Dhaka and Chittagong report the highest.

Similarly, outpatient visits are forecasted to grow by 11% in 2025 and 32% in 2026. Outpatient visits for under-five children are also expected to rise, by 6% in 2025 and 27% in 2026. All divisions show an increasing trend in outpatient service utilization, with Sylhet and Barishal recording the lowest usage and Dhaka and Chittagong the highest (Fig 3).

## Discussion

This study examines trends in health service utilization in Bangladesh and projects future utilization, highlighting the urgent need to expand the availability and capacity of healthcare facilities to meet the anticipated growth in service coverage. Forecasts indicate that the use of nearly all essential health services will continue to rise in the coming years. The most substantial projected increase is observed in the provision of Kangaroo Mother Care (KMC) for facility-born infants, expected to rise by 20% in 2025 and by 74% in 2026 compared to 2024. In contrast, the smallest projected increases are in the administration of child immunisations. The analysis also reveals significant geographic disparities in service use. Barishal and Sylhet divisions consistently showed the lowest coverage across most indicators, while Dhaka and Chittagong reported the highest levels of service utilization.

Our study identified a steady rise in the number of deliveries conducted in public health facilities across Bangladesh aligned with a previous study [21], reflecting a broader national shift toward institutional childbirth. This aligns with findings from previous national surveys, including the Bangladesh Demographic and Health Surveys (BDHS) of 2017–18 and 2022, which also reported increasing trends in facility-based births, particularly in public sector institutions [22,23]. The growing reliance on facility deliveries is likely driven by the government's continuous investments in maternal health infrastructure, increased deployment of skilled birth attendants and the integration of delivery services into primary healthcare centers [24]. In addition, awareness campaigns promoting the benefits of institutional deliveries may have played a role in influencing community behavior. In addition, Bangladesh's consistently high birth rate has increased the absolute demand for delivery services, which has led to an increase in institutional births and put extra pressure on public health facilities.

Our findings also reveal a rising trend in cesarean section (C-section) deliveries, which is consistent with earlier research showing Bangladesh's sharp increase in C-section rates in recent years. This growing rate, while partly indicative of improved access to emergency obstetric care, also raises concerns about potential over-medicalization and unnecessary surgical interventions [25,26]. Regionally, our analysis showed higher rates of C-section deliveries in Rajshahi, Mymensingh and Dhaka, mirroring trends reported in earlier studies [27]. The increased demand for maternal health services overall underscores the need to strengthen health system readiness, particularly by ensuring the availability of

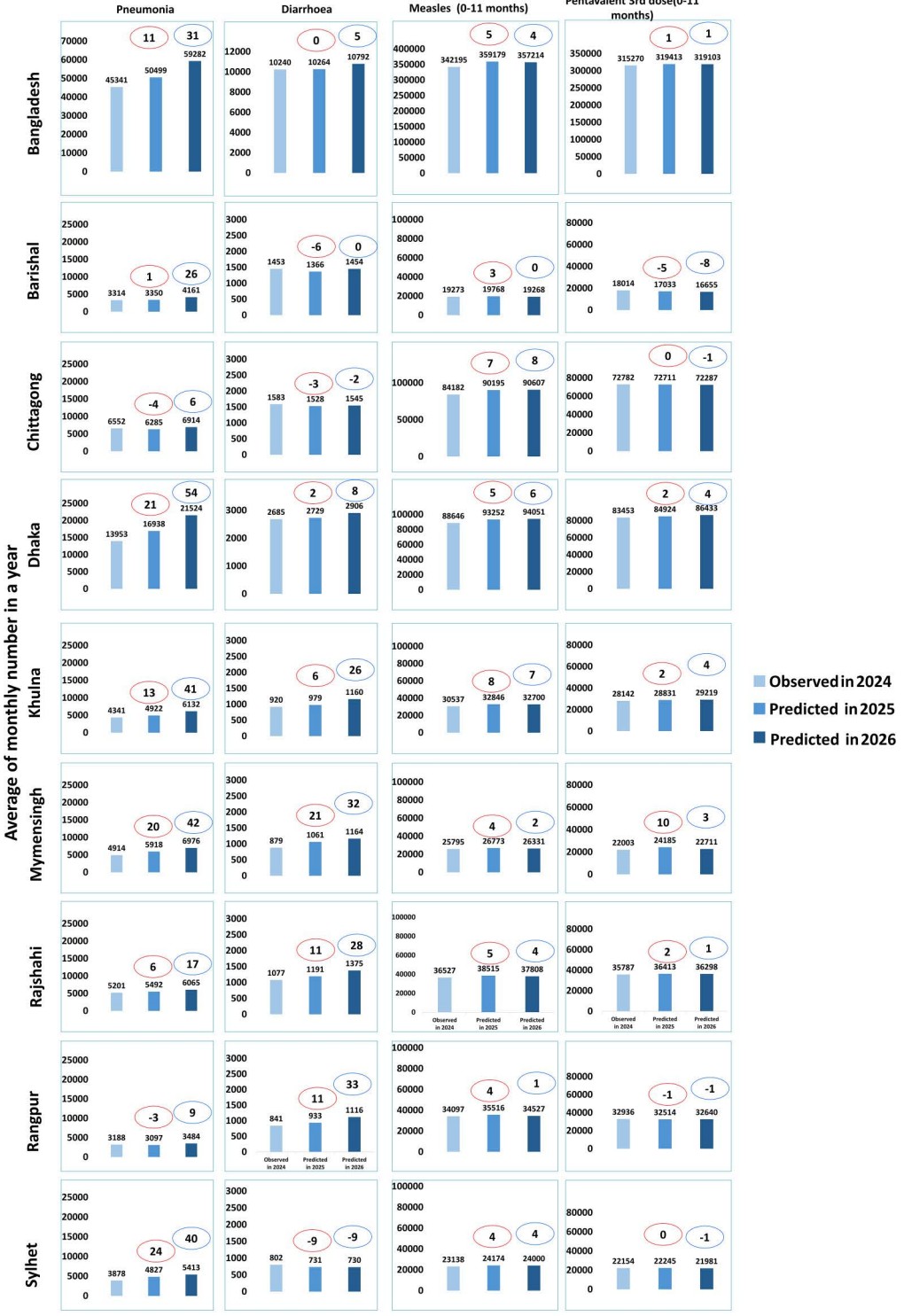

**Fig 2. Relative change in Child health service utilization during 2025 and 2026 compared to 2024.** Bars represent number of child health service utilization per year.

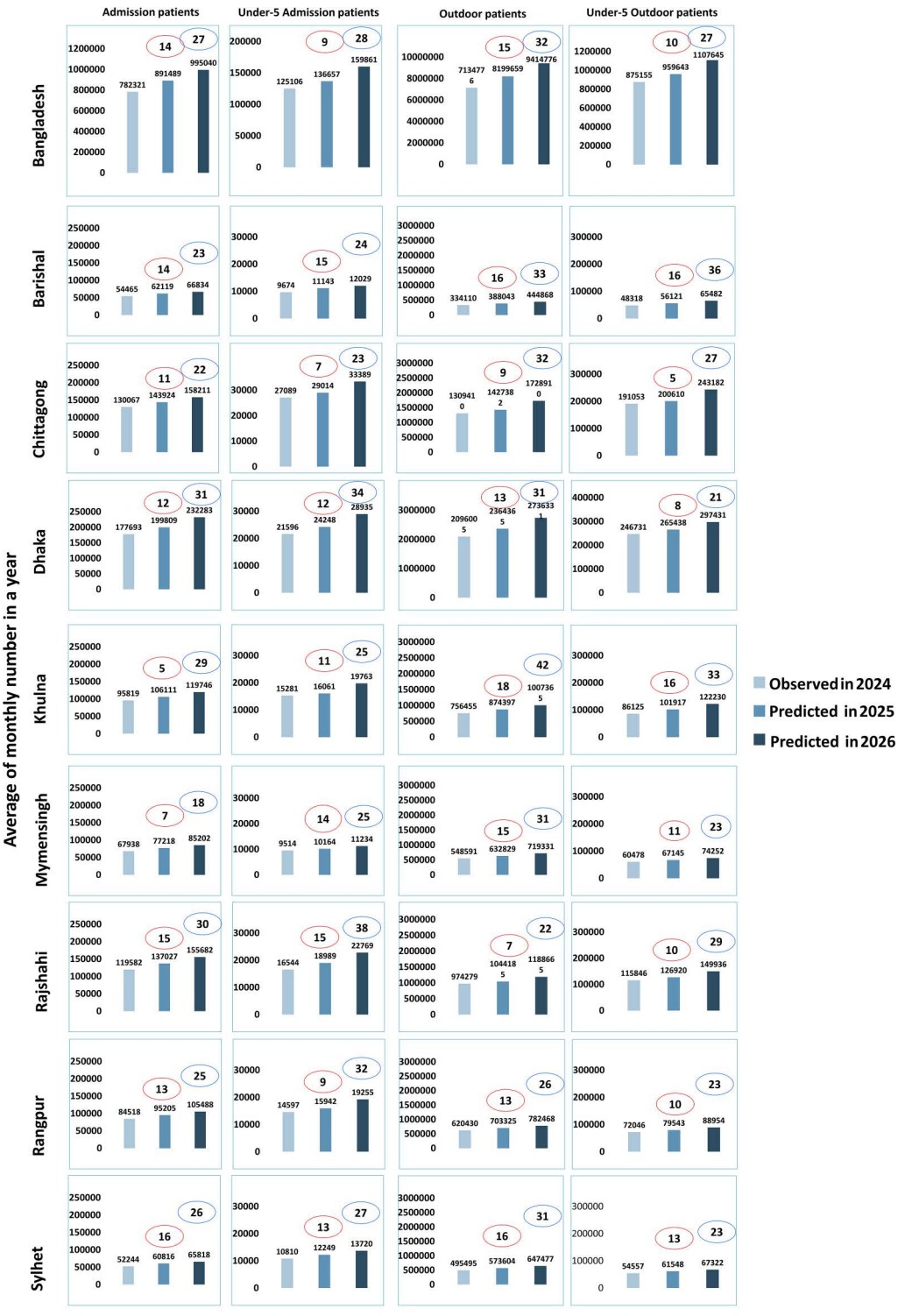

**Fig 3. Relative change in hospital visits during 2025 and 2026 compared to 2024.** Bars represent number of hospital visits per year.

trained personnel, enhancing quality of care during childbirth and implementing stricter clinical protocols to guide the use of C-sections. In Rangpur, Barishal and Khulna, normal deliveries are trending downward. While home deliveries continue to be quite common in the Barisal division [28], C-section deliveries are very common in the Khulna division [29]. Due in large part to a lack of knowledge about the dangers of needless C-sections, many families are choosing surgical procedures over natural births, leading to a decline in normal deliveries.

In addition to the overall upward trend in facility-based deliveries, our study revealed a clear seasonal pattern, with deliveries peaking between September and December. This finding is consistent with previous research indicating a strong seasonality in childbirth patterns in Bangladesh, where nearly twice as many births occur in December compared to July [30]. These seasonal peaks may reflect a combination of biological, environmental and socio-cultural factors that influence conception timing and thus delivery seasonality. The evidence of seasonality suggests the need for anticipatory health system planning, including allocating sufficient staff, beds and emergency care capacity during expected high-demand months, to ensure that the quality of care is not compromised and that medically unnecessary procedures are minimized.

Low birth weight (LBW) remains a significant concern, with our projections showing continued increases, which supports earlier findings indicating the prevalence of LBW is likewise increased in Bangladesh [31]. This is increasing particularly in Rajshahi, Dhaka and Sylhet. The rise in LBW is associated with poor maternal nutrition, inadequate antenatal care and suboptimal monitoring during pregnancy [32]. While Dhaka's urban slums face overcrowding and environmental hazards, Rajshahi and Sylhet may suffer from weaker infrastructure and limited access to skilled care. The strong seasonal peak in LBW from September to December—coinciding with high birth volumes—further underscores the need to improve maternal nutrition and antenatal support. Targeted investments in quality ANC and PNC services, especially in high-burden regions, are critical to addressing this challenge.

The increasing trend in KMC coverage is encouraging, suggesting that Bangladesh is making progress in adopting this evidence-based intervention for LBW babies [33,34]. All divisions, except Khulna, showed an upward trajectory, but gaps remain in ensuring that all eligible newborns receive KMC. These gaps could be due to infrastructure limitations, lack of trained personnel, or inconsistent adherence to protocols. Greater investment in facility readiness, health worker training and caregiver education will be essential to scale up KMC equitably. We found that the number of LBW babies has not increased as quickly as KMC coverage. This is probably due to the fact that KMC is a relatively new intervention and that there are an increasing number of facilities that offer it.

Trends in child health outcomes were mixed. Diarrhea treatment showed a modest increase, while pneumonia treatment is projected to rise sharply. This mirrors prior research that observed an initial decline in diarrhea prevalence followed by a resurgence after 2019 [35]. Additionally, our data show clear seasonal peaks, especially in October–November and March–April, which are known times when the prevalence of diarrhea is higher due to environmental and meteorological variables. A similar pattern was observed in a prior investigation [36]. In order to handle seasonal surges, facilities should have enough people and resources, especially during known peak months. Enhancing access to clean water, sanitation and hygiene (WASH), encouraging prompt care-seeking behavior and guaranteeing that oral rehydration therapy and zinc supplements are available in primary care facilities should be the main goals of public health initiatives. One of the biggest causes of illness and mortality worldwide, especially for children under five, is pneumonia. In this study, we saw a rising trend of Bangladeshi patients seeking therapy for pneumonia. It is encouraging that a decrease in pneumonia-related mortality in recent years seems to be linked to this increase in care-seeking behavior [37]. Despite these advancements, estimates suggest that if prevention and treatment measures are not strengthened, pneumonia may still kill over 100,000 children in Bangladesh under the age of five during the course of the next ten years [38]. Notably, Dhaka showed the highest number of pneumonia cases, a trend that may be attributed to the city's high levels of air pollution, population density and poor housing conditions. Dhaka consistently ranks among the most polluted cities in the world, which have been linked to increased respiratory infections in children [39]. Overcrowding in urban slums,

limited ventilation and exposure to indoor smoke may further exacerbate children's vulnerability. These findings highlight the urgent need for region-specific strategies that strengthen pneumonia prevention and treatment, particularly in urban centers like Dhaka, including promoting clean air initiatives, ensuring the availability of essential medicines and raising caregiver awareness on early care-seeking.

Our analysis shows that the number of key childhood immunizations, such as the third dose of the pentavalent vaccine (Penta-3) and the measles vaccine, continues to increase slightly across most divisions in Bangladesh. These results are consistent with earlier studies that also indicated a nationwide increase in Penta-3 coverage [40]. This slight upward trend is due to the performance of the country's Expanded Programme on Immunization (EPI), which is recognised for its success in reaching the majority of children with routine vaccines. The national immunization system has achieved consistently high coverage levels, supported by strong government leadership and regular community outreach. Therefore, the relatively small projected increases in Penta-3 and measles vaccinations number are not indicative of system limitations, but rather reflect an expected increase in the number of births and thus a greater volume of children entering the immunization schedule.

Our study projected a substantial rise in hospital admissions and outpatient visits across Bangladesh. Similar trends were observed for under-five admissions and outpatient consultations. This consistent upward trajectory suggests a growing demand for facility-based care, likely driven by multiple factors including population growth, greater awareness of health services, improved access through public sector expansion and a rebound in healthcare-seeking behavior following the COVID-19 pandemic [41].

The high predicted numbers have clear implications for health system planning. The healthcare system remains constrained by limited bed capacity, staff shortages, and inadequate infrastructure. To ensure timely, equitable and quality care, it is essential to strengthen hospital readiness by expanding physical infrastructure, increasing the healthcare workforce and improving the availability of diagnostics and essential supplies. Special attention should also be paid to scaling up services during seasonal peaks through flexible staffing and surge capacity planning. These actions will be critical to ensuring that Bangladesh's health system can keep pace with rising service demands and provide efficient, people-centered care across all divisions.

## Implications for health service planning

The findings of this study underscore the urgent need for proactive and data-driven health system planning in Bangladesh. With projected increases in maternal, newborn, child and hospital-based service utilization, particularly in densely populated divisions like Dhaka and Chittagong, health authorities must invest in strengthening infrastructure, workforce capacity and service delivery readiness. The substantial rise in utilization of Kangaroo Mother Care, pneumonia treatment and facility-based deliveries highlights the need for targeted resource allocation, especially in underperforming regions such as Sylhet and Barishal. Routine health information platforms like DHIS2 can play a transformative role by enabling near real-time monitoring and forecasting of service needs. Integrating such forecasting tools into routine health planning at national and subnational levels will support timely decision-making, improve equity in resource distribution and enhance the overall resilience of Bangladesh's health system.

## Strength and limitations

Our study offers a comprehensive analysis of health service utilization trends in Bangladesh using a large dataset from DHIS2, which is widely recognised for its structured and consistent reporting system. By incorporating over four years of monthly data (2021–2025), the study captures both short- and long-term fluctuations in service use, allowing us to detect seasonal patterns and regional variations. Our use of a Bayesian time-series approach with a log-linear Poisson model and adjustment for seasonality and autocorrelation enhances the methodological rigor of our forecasts. Furthermore, our divisional-level projections highlight important regional disparities and offer timely, actionable insights to support localized health system preparedness and planning efforts.

PLOS Global Public Health

However, like most studies relying on routine health information systems, our analysis has limitations. First, DHIS2 data are aggregated at the facility level, limiting our ability to explore individual-level determinants such as age, sex, socioeconomic status, or household characteristics. This prevents us from examining the role of individual risk factors or behaviors in shaping service utilization. Second, although DHIS2 is a robust platform, data quality and completeness may vary across regions and over time. For example, incomplete data entry, skipped indicators and reporting delays could have led to under- or overestimation of health service use. Lastly, our dataset includes mostly public health facilities; majority of private sector service use is not captured, despite its substantial role in the country's healthcare landscape [20]. This may result in an incomplete picture of overall service demand, particularly in urban areas where private health services are more commonly accessed. Service utilization is also driven by governments and partners' activities to increase demand for and access to services. Past trends may be driven by such activities. Our projections are conservative and assume that similar activities will continue linearly in the future. This may not be necessarily the case. The analysis also assumed continue linear growth in the population based on historical trends.

Despite these limitations, our findings underscore the value of routine data systems like DHIS2 in generating timely, policy-relevant insights and demonstrate the feasibility of applying predictive analytics for health system strengthening in resource-constrained settings. We used time series components like trend and seasonality in our model. Other possible factors were not included in the model, such as birth rate, population growth, initiatives from government and demographic shifts like aging. Therefore, our forecasts might not accurately reflect future changes if additional government actions are implemented.

## Conclusion

This study presents the first national-level attempt to forecast health service demand in Bangladesh using routine DHIS2 data. Our findings highlight a steady and significant increase in the utilization of maternal, newborn, child and hospital-based health services across the country, with clear seasonal and regional patterns. These projections have critical implications for health system planning, particularly in a resource-constrained context like Bangladesh, where service delivery is already strained by workforce shortages, infrastructure limitations and inequities in access. The sharp projected increases in services such as Kangaroo Mother Care, pneumonia treatment and hospital admissions signal growing pressure on an already fragile health system. Regional inequalities indicate the necessity for geographically specific resource allocation strategies. These include the persistently low service coverage during daily hospital visits and the admission data in Sylhet and Barishal compared to high utilization in Dhaka and Chittagong. To ensure readiness and equity, Bangladesh must invest in strengthening health system capacity through strategic workforce deployment, infrastructure expansion and the use of data-driven planning tools. Leveraging platforms like DHIS2 for real time forecasting and operational decision-making can transform the country's ability to respond to rising and shifting healthcare needs. As Bangladesh works toward achieving Universal Health Coverage and the Sustainable Development Goals, these insights offer a roadmap for building a more resilient, responsive and equitable health system.

## Supporting information

**S1 File. Trend and prediction of maternal and neonatal health services.**
(PDF)

**S2 File. Trend and prediction of child health services.**
(PDF)

**S3 File. Trend and prediction of hospital-based services.**
(PDF)

**S4 File. Posterior predictive check comparing observed data with 95% predictive credible intervals.**
(PDF)

**S5 File. Description of study indicators.**
(PDF)

## Acknowledgments

We gratefully recognize the Management Information System, Directorate General of Health Services, Bangladesh. We also acknowledge the essential technical partnership and guidance provided by Johns Hopkins University. icddr,b appreciatively acknowledges its core donors for their ongoing commitment and support.

## Author contributions

**Conceptualization:** Toufiq Hassan Shawon, Ema Akter, Ahmed Ehsanur Rahman, Shams El Arifeen, Agbessi Amouzou, Aniqa Tasnim Hossain.

**Data curation:** Toufiq Hassan Shawon, Ema Akter, Bibek Ahamed, Aniqa Tasnim Hossain.

**Formal analysis:** Ema Akter, Bibek Ahamed.

**Investigation:** Ahmed Ehsanur Rahman, Shams El Arifeen, Agbessi Amouzou, Aniqa Tasnim Hossain.

**Methodology:** Ema Akter.

**Software:** Ema Akter.

**Supervision:** Shams El Arifeen, Agbessi Amouzou, Aniqa Tasnim Hossain.

**Validation:** Tazreen Ahmed, Romana Afroz Lubna, Animesh Biswas, Tasnu Ara, Ridwana Maher Manna, Pradip Chandra, Nasimul Gani Usmani, Md. Alamgir Hossain, S.M. Hasibul Islam, Sultan Mahmud, Abu Bakkar Siddique, Shafiqul Ameen, Shabnam Mostari, Mohammad Sohel Shomik, Emily Wilson, Nadia Akseer.

**Visualization:** Bibek Ahamed.

**Writing – original draft:** Toufiq Hassan Shawon, Ema Akter, Bibek Ahamed.

**Writing – review & editing:** Tazreen Ahmed, Romana Afroz Lubna, Animesh Biswas, Tasnu Ara, Ridwana Maher Manna, Pradip Chandra, Nasimul Gani Usmani, Md. Alamgir Hossain, Md. Shahidul Islam, S.M. Hasibul Islam, Sultan Mahmud, Abu Bakkar Siddique, Shafiqul Ameen, Shabnam Mostari, Mohammad Sohel Shomik, Emily Wilson, Nadia Akseer, Ahmed Ehsanur Rahman, Shams El Arifeen, Agbessi Amouzou, Aniqa Tasnim Hossain.

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
