## [Decision Letter · Decision Letter 0]

28 Dec 2025

PGPH-D-25-02646

Using DHIS2 routine data for health system preparedness in resource-limited settings: A bayesian predictive approach in Bangladesh

Dear Dr. Hossain,

Thank you for submitting your manuscript to PLOS Global Public Health. After careful consideration, we feel that it has merit but does not fully meet PLOS Global Public Health’s publication criteria as it currently stands. Therefore, we invite you to submit a revised version of the manuscript that addresses the points raised during the review process.

We look forward to receiving your revised manuscript.

Kind regards,

Saifur R. Chowdhury, BScN, MPH, PhD (c)

Academic Editor

Journal Requirements:

Additional Editor Comments (if provided):

Reviewers' comments:

Reviewer's Responses to Questions

**Comments to the Author**

1. Does this manuscript meet PLOS Global Public Health’s publication criteria ? Is the manuscript technically sound, and do the data support the conclusions? The manuscript must describe methodologically and ethically rigorous research with conclusions that are appropriately drawn based on the data presented.

Reviewer #1: Yes

Reviewer #2: Yes

Reviewer #3: Yes

Reviewer #4: Yes

2. Has the statistical analysis been performed appropriately and rigorously?

Reviewer #1: No

Reviewer #2: Yes

Reviewer #3: Yes

Reviewer #4: Yes

3. Have the authors made all data underlying the findings in their manuscript fully available (please refer to the Data Availability Statement at the start of the manuscript PDF file)?

Reviewer #1: No

Reviewer #2: Yes

Reviewer #3: No

Reviewer #4: Yes

4. Is the manuscript presented in an intelligible fashion and written in standard English?

Reviewer #1: Yes

Reviewer #2: Yes

Reviewer #3: Yes

Reviewer #4: Yes

5. Review Comments to the Author

Reviewer #1: The manuscript applies accurate statistical frameworks, however, the statistical section lacks sufficient detail and transparency for reproducibility and proper evaluation. The manuscript does not provide adequate information on model specification, prior selection, model convergence, diagnostics, or validation procedures. It is unclear how the authors addressed potential overdispersion or autocorrelation beyond first order, and no measures of model performance are reported.

I recommend the authors:

Include a clear description of priors, hyperparameters, and assumptions.

Provide diagnostic plots and model fit indices in supplementary materials.

Demonstrate the robustness of forecasts through sensitivity analyses or model comparison.

Best of luck with your revisions :) I may not be right, but please do pay attention to these points.

Reviewer #2: It is an excellent work where researchers used DHIS 2 platform to have an idea on trend of service utilisation of maternal , neonatal and child health, vaccination status especially penta 3 and measles. It also explored disparities of service utilization and delivery between different divisions of Bangladesh and seasonal variations as well. This work also showed projection of different services in public health facilities . Such evidences will help policy makers for future planning , priority setting as well. Although having some limitation it might have impact in planning to handle crisis.

Despite having good quality of this manuscript information regarding IRB clearance / permission from appropriate authority seems missing.

Reviewer #3: This manuscript, “Using DHIS2 routine data for health system preparedness in resource-limited settings: A Bayesian predictive approach in Bangladesh”, presents a well-designed and policy-relevant study applying Bayesian log-linear Poisson regression to forecast service utilization trends using DHIS2 data from Bangladesh. The study is technically rigorous, original, and offers important implications for data-driven health system planning and preparedness in LMICs.

The topic aligns strongly with PLOS Global Public Health’s scope, focusing on strengthening health systems and improving resource allocation through innovative data use. The paper is well-structured and methodologically sound, and the conclusions are appropriately supported by the data. The authors provide clear evidence of rising health service utilization and highlight regional disparities, which are both timely and significant findings for the Bangladeshi health system.

The following minor revisions are suggested primarily to improve clarity, reporting precision, and conciseness.

1. Abstract: The abstract is informative but overly detailed with repeated numeric projections.

Suggested Action: Summarise the main findings more concisely and reduce numeric repetition.

2. Methods (Model Validation): The analytical approach is well-explained, but details on model validation or diagnostic checks are missing.

Suggested Action: Briefly describe how model fit or predictive performance was assessed (e.g., residual checks, posterior predictive accuracy, etc).

3. Data Availability Statement: The data are restricted under DGHS permission. While this is reasonable, the statement could more clearly justify restricted access.

Suggesred Action: Expand the justification for restricted access and outline how data can be requested formally by indicate contact address. Alternatively deposit anonymised data in a public repository.

4. Language and Clarity: The manuscript is well-written overall but includes some long sentences and minor grammatical inconsistencies.

Suggested Action: Edit for brevity and clarity. Consider light professional copy editing.

i) Examples of Long or Complex Sentences

a). Page 2, lines 33-38 (Abstract) “By analysing service utilization trends and projecting future service volume at national and regional level, we aim to improve region-specific health planning and promote more efficient and equitable health service provision.”

Suggested Edit: Long but clear; could be split after “level.” → “By analysing service utilization trends and projecting future service volume at national and regional levels, we aim to improve region-specific health planning. This can promote more efficient and equitable service provision.”

b) Page 6, lines 120-126 “Resource allocation decisions in this context are frequently made without sufficient data-driven insights, leading to inefficiencies, supply shortages, and gaps in service delivery.”

Grammatically correct but wordy; could shorten to: “Resource allocation decisions often lack data-driven insights, causing inefficiencies and service gaps.”

c) Page 7, lines 155-162 “Routine health information systems, such as the District Health Information Software 2 (DHIS2), offer a promising opportunity to strengthen health system preparedness and planning (16). As a widely implemented digital platform for collecting, managing, and analyzing health data, DHIS2 has been adopted in over 60 countries and serves as a backbone for health service reporting in Bangladesh.”

These two sentences are dense and repetitive; merging or trimming one clause would improve flow.

d) Page 10, lines 270-278 “We employed a Bayesian approach to forecast service utilisation. As recommended by earlier research, a log-linear model was used to fit the model and forecast future health services, assuming that the number of persons using health services at the 't' time point follows a Poisson distribution (20).”

Repetition of “model”; could simplify to: “Following prior research, we used a Bayesian log-linear Poisson model to forecast future health service use.”

e) Page 17, lines 535-542 (Discussion) “In addition to the overall upward trend in facility-based deliveries, our study revealed a clear seasonal pattern, with deliveries peaking between September and December. This finding is consistent with previous research indicating a strong seasonality in childbirth patterns in Bangladesh, where nearly twice as many births occur in December compared to July (30).”

Long but grammatically fine; may read better if split after “December.”

f) Page 20, lines 640-647 “Without proactive investments, existing health facilities may become overstretched, especially in high-demand regions like Dhaka and Chattogram. Bangladesh’s healthcare system continues to face challenges related to limited bed capacity, shortages of trained personnel, and inadequate infrastructure, particularly in peripheral and rural areas.”

Sentence length acceptable but repetitive phrase “continues to face challenges related to” could be simplified: “The healthcare system remains constrained by limited bed capacity, staff shortages, and inadequate infrastructure.”

ii) Examples of Minor Grammatical or Stylistic Inconsistencies.

a) Page 5, line 102 “The timely delivery of treatment is directly affected by the efficient use of medical resources, which is an increasing problem for healthcare services (2).”

Issue / Suggestion: Awkward phrasing; consider “inefficient use … is an increasing problem.”

b) Page 7, line 148 “The country is organized into eight major administrative divisions: Barishal, Chattogram, Dhaka, Khulna, Rajshahi, Rangpur, Mymensingh, and Sylhet.”

Issue / Suggestion: Consistent; no issue—listed here because commas occasionally vary elsewhere (“and”)—ensure consistent punctuation style.

c) Page 9, line 220 “During regular service delivery, healthcare practitioners document health services in real time, which are then uploaded daily and some cases monthly to the DHIS2 platform and recorded into facility registers (19).”

Issue / Suggestion: Missing preposition: “in some cases monthly.”

d) Page 11, line 310 “Projections indicate that the number of normal deliveries will rise by approximately 6% in 2025 and 10% in 2026 compared to 2024 (Fig 1).”

Issue / Suggestion: Numerically correct but check for consistent use of “%” symbol (some later sentences spell out “percent”).

e) Page 18, line 585 “Due in large part to a lack of knowledge about the dangers of needless C-sections, many families are choosing surgical procedures over natural births, which is why the number of normal deliveries is declining.”

Issue / Suggestion: Stylistic: avoid causal phrase “which is why” in formal academic writing → “leading to a decline in normal deliveries.”

f) Page 23, line 730 “Service utilisation is also driven by governments and partners’ activities to increase demand of and access to services.”

Issue / Suggestion: Slightly awkward phrase “demand of” → should be “demand for.”

g) Page 25, line 805 “These include the persistently low service coverage during daily hospital visits and the admission data set in Sylhet and Barisal compared to high utilization in Dhaka and Chattogram.”

Issue / Suggestion: “Admission data set” unclear; perhaps “admission data show low service coverage.”

h) Page 26, line 820 “Leveraging platforms like DHIS2 for real-time forecasting and operational decision-making can transform the country’s ability to respond to rising and shifting healthcare needs.”

Issue / Suggestion: Fine but ensure hyphen consistency (“real-time” used elsewhere as “real time”).

5. References: Minor inconsistencies in citation formatting (capitalization, punctuation).

Suggested Action: Standardize all references per PLOS Global Public Health guidelines.

6. Figures and Supplementary Files: Figures are clear and relevant, but figure legends could be slightly expanded for standalone interpretation.

Suggested Action: Ensure legends briefly summarize key findings for each figure.

Reviewer #4: The manuscript addresses a very important and novel topic using DHIS2 routine data to assess maternal, child, and newborn health indicators in Bangladesh. The seasonal patterns in child health outcomes are highlighted well, and the indicators selected are appropriate.

Methods: Data cleaning procedures are unclear, including handling of missing values or inconsistencies. Please describe how missing data were managed. Consider a table summarizing all indicators, definitions, and units for clarity.

Analysis: The approach for modeling seasonality is not clearly described. Please clarify whether Fourier terms, dummy variables, or other methods were used.

Results/Figures: In Figure 1, the colors for “observed 2024” and “predicted 2025” are too similar; use more distinct colors for readability.

Language and Formatting: Typo in Discussion: “utlisation” instead of “utilization.” Some references are old, and journal formatting is inconsistent. The manuscript uses “Chittagong” and “Chattogram” interchangeably, clarify that they refer to the same division.

No concerns regarding dual publication, research ethics, or publication ethics were identified. Minor edits and clarifications will improve clarity and rigor.

6. PLOS authors have the option to publish the peer review history of their article (what does this mean? ). If published, this will include your full peer review and any attached files.

**Do you want your identity to be public for this peer review?** For information about this choice, including consent withdrawal, please see our Privacy Policy .

Reviewer #1: **Yes:** Hana Abbasian

Reviewer #2: No

Reviewer #3: No

Reviewer #4: **Yes:** Syed Muhammad Ali Abidi

Figure Resubmissions:

---

## [Editor Report · Decision Letter 1]

4 Feb 2026

Using DHIS2 routine data for health system preparedness in resource-limited settings: A bayesian predictive approach in Bangladesh

PGPH-D-25-02646R1

Dear Dr. Hossain,

We are pleased to inform you that your manuscript 'Using DHIS2 routine data for health system preparedness in resource-limited settings: A bayesian predictive approach in Bangladesh' has been provisionally accepted for publication in PLOS Global Public Health.

Best regards,

Saifur R. Chowdhury, BScN, MPH, PhD (c)

Academic Editor